# The chromosome-scale genome sequence of *Triadica sebifera* provides insight into fatty acids and anthocyanin biosynthesis

Jie Luo[1], Wenyu Ren[1], Guanghua Cai[1], Liyu Huang[1], Xin Shen[2], Na Li[3], Chaoren Nie[3], Yingang Li [2✉] & Nian Wang [1✉]

The Chinese tallow tree (*Triadica sebifera*) can produce oil with high content of unsaturated fatty acids in seeds and shows attractive leaf color in autumn and winter. Here, the 739 Mb chromosome-scale genome sequence of the Chinese tallow tree was assembled and it reveals the Chinese tallow tree is a tetraploid. Numerous genes related to nutrition assimilation, energy utilization, biosynthesis of secondary metabolites and resistance significantly expanded or are specific to the Chinese tallow tree. These genes would enable the Chinese tallow tree to obtain high adaptability. More genes in fatty acids biosynthesis in its genome, especially for unsaturated fatty acids biosynthesis, and higher expression of these genes in seeds would be attributed to its high content of unsaturated fatty acids. Cyanidin 3-*O*-glucoside was identified as the major component of anthocyanin in red leaves. All structural genes in anthocyanin biosynthesis show significantly higher expression in red leaves than in green leaves. Transcription factors, seven MYB and one bHLH, were predicted to regulate these anthocyanin biosynthesis genes. Collectively, we provided insight into the polyploidization, high adaptability and biosynthesis of the high content of unsaturated fatty acids in seeds and anthocyanin in leaves for the Chinese tallow tree.

[1] College of Horticulture and Forestry Sciences, Huazhong Agricultural University, Wuhan 430070, China. [2] Forest Breeding Institute, Zhejiang Academy of Forestry, Hangzhou 310023, China. [3] Wuhan Institute of Landscape Architecture, Wuhan 430070, China. ✉email: hzliyg@126.com; wangn@mail.hzau.edu.cn

In the spurge family (Euphorbiaceae), there are ca. 300 genera and 8000 species. Numerous plants in this family can be used for generating bioenergy and bioproducts. For example, the rubber tree (*Hevea brasiliensis*) produces a commercially viable amount of natural rubber, and castor bean (*Ricinus ommunis*) is used to make castor oil, which is a strong laxative. Although plants in the spurge family are important to us, there is a lack of understanding of these plants, especially regarding their genomic and genetic information. Recently, the genome sequences of some plants in the spurge family, including the rubber tree, castor bean, physic nut (*Jatropha curcas*), tung tree (*Vernicia fordii*), and cassava (*Manihot esculenta*), were released[1–5]. The unigenes of leafy spurge (*Euphorbia esula*) is a perennial weed in the spurge family, was also released with transcriptome assembly[6]. The Chinese tallow tree (CTT) is also an important species in the spurge family and is native to China. The previous Latin name of this plant was *Sapium sebiferum*, and this name was recently changed to *Triadica sebifera* (http://floranorthamerica.org/Triadica_sebifera). The CTT has been cultivated in China for at least 1500 years. Seeds of the CTT can produce up to ca. 40% oil[7]; thus, the CTT is also an important energy oil plant in the spurge family. However, even as knowledge has been obtained, there is no genome resource for this important tree species.

Seeds of the CTT can produce two types of oil, stillingia oil and tallowy fat[8,9]. Both types of oil are important resources for the food industry and biofuel production. The tallow fat is produced in the white coat of the seeds, while the stillingia oil accumulates in the seed kernels. Moreover, there is a relatively high content of unsaturated fatty acids (FAs), of which ca. 30% is linoleic acid (C18:2) and 50% is linolenic acid (C18:3)[9,10].

In addition to being used for bioenergy production, the CTT is also considered an ornamental tree in China because of its attractive leaf color in autumn and winter. In the central regions of China, the CTT is widely grown, especially in rural areas that comprise mainly mountains. The red, purple, and yellow colors of the CTT leaves attract much attention in autumn and winter. Now there are ca. 20 cultivars in China used as ornamental trees, such as 'Hongzi Jiaren', 'Pudazi', and 'Huihuang' (Personal communication). Therefore, the CTT has become an important tourism resource in some places in the central regions of China, e.g., Luotian and Dawu Counties of Hubei province. In recent years, some elite CTT lines with attractive leaf colors have been introduced to some large cities in China for landscaping. Additionally, the CTT also produces some secondary metabolites that can be used as medicine. For example, nine compounds, shikimic acid, kaempferol, quercetin, isoquercein, hyperin, kaempferol-3-O-beta-D-glucopyranoside, kaempferol-3-O-beta-D-glueopyranoside, gallic acid, and rutin, were extracted from leaves of the CTT and these chemical compounds are the compositions of some herbal medicines[11]. Though the CTT is an important tree with high economic value, the understanding of this plant is very limited, possibly due to the lack of genomic resources.

The CTT is also an important plant for ecological study. This tree has high adaptability and grows fast and well under various conditions, including canal and stream banks, steep mountain slopes, sandy beaches, and alkaline, saline, or acid soils. Some lines were introduced into the United States of America (USA) in the 18th century and then spread rapidly[12]. Now the CTT is considered a serious invader in the southeastern USA that is overgrowing native plants[13]. Some studies have uncovered the underlying mechanisms for its invasive characteristics. For example, a higher flavonoid concentration in the roots and higher concentration of flavonoids, and lower concentration of tannins were found in introduced CTTs than in native plant populations[14–16]. However, there are no reports on genomic levels to reveal its invasive mechanisms.

To meet the requirement of high-quality genomic resources of CTT, we assembled a chromosome-scale genome sequence of the CTT. The genome evolution of CTT and some species in the spurge family were analyzed. The cause of the high adaptability of CTT was predicted by a comprehensive investigation of the features of its genome. We also examined the compositions of anthocyanin in red CTT leaves. The biosynthesis pathways for the formation of FAs in seeds and anthocyanin in red leaves were also dissected with the CTT genome sequence. In summary, we obtained a valuable genome resource for the CTT and provided insight into polyploidization, high adaptability, and biosynthesis of the high content of unsaturated FAs in seeds and anthocyanin in leaves for this tree plant.

## Results

**Chromosome-scale genome of the Chinese tallow tree (CTT) revealed that it is a tetraploid.** The CTT line used for genome assembly showed red leaves in the field in Luotian County, Hubei Province, China (N 31.05, E 115. 66, H 387.7 m) (Fig. 1a, c). Its female flowers grow in the lower part of the inflorescence, while the male flowers grow in the upper part (Fig. 1b). Seeds of the CTT are coated with white wax (Fig. 1d). To estimate the genome size of the CTT, 104.86 Gb PE reads were used to perform 17-bp K-mer analysis (Supplementary Data 1). Interestingly, a total of 3 peaks can be observed for the K-mer distribution (Supplementary Fig. 1). The distribution of K-mer frequency suggests that the genome of the CTT may be tetraploid and that the tetraploid genome size of the CTT is 2.95 Gb. Additionally, there was also an increased frequency after 100-fold coverage, suggesting a high percentage of repeated sequences in the CTT genome.

The predicted percent of repeats in the genome was 64.80%. Subsequently, the genome size of another two CTT lines was also estimated by flow cytometry. By setting the diploid genome size of poplar as 912 Mb[17], the genome sizes of the two CTT lines were estimated as 2934 and 3010 Mb (Supplementary Data 2). Therefore, the two estimations were almost the same.

To assemble the CTT genome, a total of 42.81 Gb clean HiFi long reads were generated (Supplementary Data 1). The genome was then assembled by using two different software packages, Canu and HifiAsm. Both software programs produced a total of ca. 2.9 Gb contigs (Supplementary Data 3), while haplotype assembly was also produced by HifiAsm. Both haplotypes produced by HifiAsam were ca. 1.4 Gb (Supplementary Data 3). Moreover, the 2.9 Gb contigs produced by Canu were also purged, and the resulting size was ca. 787 Mb (Supplementary Data 3). These results also suggested that the genome of the CTT is tetraploid. Therefore, the 2.9 Gb contigs assembled by Canu (assigned as tetraploid), one of the 1.4 Gb haplotype genome produced by HifiAsm (assigned as diploid), and 787 Mb purged contigs (assigned as monoploid) were selected for further analyses. The size of N50 for the diploid and monoploid contigs were 8.4 and 29.4 Mb (Supplementary Data 3), respectively. Detailed information on the three types of data is listed in Supplementary Data 3. The three types of assemblies were first assessed for their genome completeness by BUSCO. Clearly, all three datasets showed high completeness (Fig. 1e); however, both tetraploid and diploid genomes showed >85% complete and duplicated BUSCOs, while the monoploid genome showed 77.2% and 20.5% complete and single-copy and complete and duplicated BUSCOs, respectively. These results also indicated that the genome of the CTT is tetraploid.

Second, all three types of assemblies were then anchored by using 127.09 Gb of Hi-C data. Unfortunately, only the monoploids were successfully clustered into large scaffolds. The resulting scaffolds were also subjected to a one-by-one alignment,

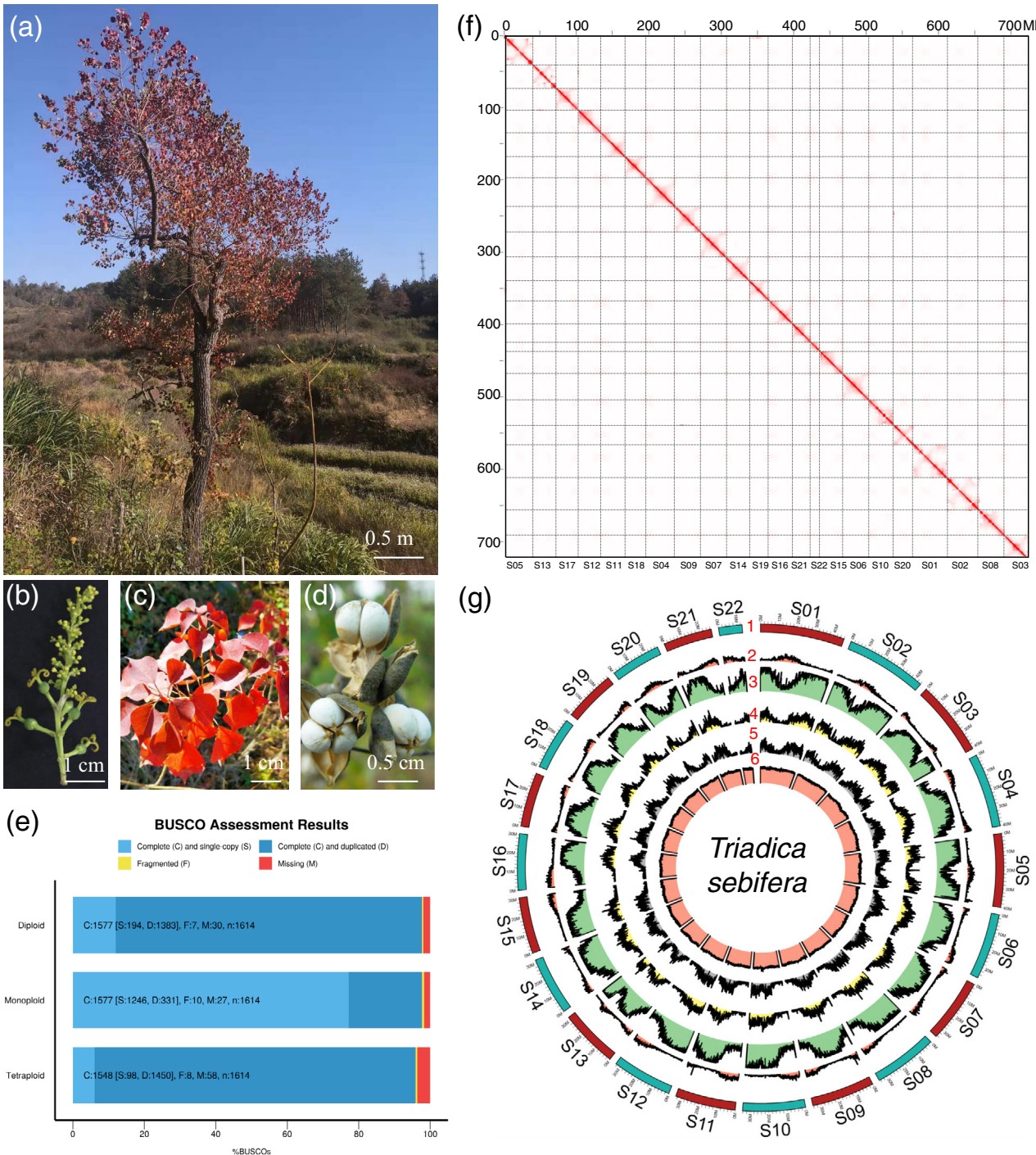

**Fig. 1 Genome assembly of the Chinese tallow tree (CTT). a** The line used as plant material in this study grows in Luotian County, Hubei Province, China. **b–d** show magnified images of fluorescence, red leaves, and seeds, respectively. **e** BUSCO assessment of the monoploid, diploid and tetraploid assemblies. **f** Plot of the Hi-C matrix for the chromosome-scale monoploid genome of the CTT. The chromosome orders for the x and y axis are the same and they are labeled in the bottom of this figure. **g** Genome features of the chromosome-scale monoploid genome of the CTT. Numbers 1 to 6 represent scaffolds (pseudochromosomes), gene density, repeat content, Gypsy transposon content, Copia transposon content, and GC content, respectively. Photos in this figure are originals and they were taken by the corresponding author Dr. Nian Wang.

and some small scaffolds that were totally covered by larger scaffolds were removed. We finally obtained 25 scaffolds with a total size of 739 Mb. A total of 22 scaffolds were >10 Mb in size, and the Hi-C matrix plot for these scaffolds is shown in Fig. 1f. A previous report suggested that there might be 88 chromosomes in the genome of the CTT (http://floranorthamerica.org/Triadica_sebifera); thus, our result supported this prediction.

Because there were no previous nomenclature of chromosomes for the CTT, we assigned these scaffolds according to their sizes, and this information is listed in Supplementary Data 4. To evaluate the monoploid genome sequence, a total of 18 RNA-Seq data were mapped onto this genome. The results showed that most RNA-Seq data had an average mapping rate of 94.28% and that most mapping rates were >90% (Supplementary Data 5).

**Table 1 Statistics of the Chinese tallow tree (CTT) genome assembly and annotation.**

| Features | Genome assembly | | |
| --- | --- | --- | --- |
| | Number | Size/mean size (bp) | |
| Raw assembly | 897 | 786,950,343 | |
| Contig N50 Length | | 29,398,129 | |
| Shortest sequence length | | 12,575 | |
| Longest sequence length | | 40,396,491 | |
| Pseudochromosomes | 22 | 739,402,270 | |
| Gene | 32,579 | 6240 | |
| Transcript | 43,536 | 2097 | |
| Coding DNA sequence (CDS) | 43,536 | 1317 | |
| Exon | 273,019 | 334 | |
| Intron | 229,483 | 808 | |
| | Repeat sequence | | |
| Type | Number | Length (bp) | Percent (%) |
| LINEs | 7337 | 3,497,379 | 0.47 |
| LTR elements | 277,801 | 365,708,966 | 49.46 |
| DNA transposons | 18,037 | 15,691,900 | 2.12 |
| Rolling-circles | 1692 | 1,376,237 | 0.19 |
| Unclassified | 225,792 | 78,602,547 | 10.63 |
| Low complexity | 36,541 | 1,790,042 | 0.24 |
| Simple repeats | 193,233 | 8,623,656 | 1.17 |
| Total | 760,433 | 475,290,727 | 64.28 |

This result suggested that the quality of the monoploid genome sequence was sufficient for further analysis. In conclusion, we generated a chromosome-scale monoploid genome sequence for the tetraploid CTT genome.

**Genome annotation.** When comprehensively investigating the 739 Mb monoploid genome sequence, the GC level was 32.21%. The repeat sequence of the monoploid genome was then identified; in total, 64.81% of the sequences were annotated as repeats (Table 1). This result highly agrees with the 17-bp K-mer estimation (Supplementary Fig. 1). Of these 64.81% repeated sequences, LTRs accounted for a large proportion (49.46%) (Table 1). In the Euphorbiaceae family, LTRs account for 14.4, 11, and 65.9% of the whole genomic sequence in *R. communis*, *Manihot esculenta*, and *H. brasiliensis*, respectively[2–4]. This result suggested that the proportion of LTR sequences varies greatly among different species in the Euphorbiaceae family.

The repeat-masked monoploid genome sequence was then subjected to gene prediction with de novo, homology, and transcript-based approaches. In total, 32,579 genes encoding 43,536 proteins were predicted with average lengths of 6240 and 1317 bp for genes and coding DNA sequences (CDSs) (Table 1), respectively. The genome features, including the distribution of GC content, repeat sequences, and genes, are illustrated in Table 1 and Fig. 1g. All these proteins were then functionally annotated with the Nr, TAIR, Swiss-Prot, KEGG, and InterPro databases. A total of 40,654 out of 43,536 proteins taking 93.4% were assigned to at least one functional term (Supplementary Data 6 and 7).

Gene prediction of the diploid assembly of CTT was also conducted with identical gene prediction pipelines to the monoploid genome and a total of 63,207 genes were predicted. Genes of monoploid and diploid genomes were used for gene cluster analysis. In total, 25,976 orthogroups were identified for these two sets of genes and the ratios of genes in diploid to monoploid were calculated for each orthogroups. Diploid gene numbers in the orthogroups that had values <1, 1 to 2 (include 1), 2, 2 to 3 (include 3), 3 to 4, 4 to 5, and >5 were 971, 14,853,

28,082, 10,090, 3048, 865, and 976 (Supplementary Data 8), respectively. There were 4142 genes taking 6.6% of genes in the diploid showed no counterparts in the monoploid genome (Supplementary Data 8). This result revealed that 6.6% of gene information was lost in the monoploid genome. These results also suggested that the CTT monoploid genome could represent most of the gene information for the diploid genome.

**Genome expansion and evolution.** To investigate the expanded and contracted genes in the genome of the CTT, the genomes of another 11 plants, namely, *A. thaliana*, *Eucalyptus grandis*, *H. brasiliensis*, *J. curcas*, *Linum usitatissimum*, *Manihot esculenta*, *Oryza sativa*, *P. trichocarpa*, *R. communis*, *Salix purpurea*, and *V. vinifera* were used for gene cluster analysis. In total, 63,750 orthogroups were identified for all 12 plants. Of these 63,750 orthogroups, 8580 were common for all 12 plants (Fig. 2a). A total of 398 orthogroups were only found in the CTT genome. These 398 orthogroups including 1781 genes, is considered to be specific for the CTT (Supplementary Data 9). Functional analysis of these 1781 genes showed enrichment in a number of biological processes, such as organonitrogen compound metabolic process, phosphorus metabolic process, carbohydrate metabolic process, organic acid metabolic process, and hydroxy compound metabolic process (Fig. 2b). Clearly, all these biological processes enable plants to obtain more energy and nutrition.

Of the 8580 orthogroups, we identified 151 orthogroups with single-copy genes. A phylogenetic tree was constructed by using these 151 single-copy genes in the 12 plants. Clearly, the five plants in the Euphorbiaceae family are clustered together (Fig. 2c). Moreover, the CTT shows the closest distance to *R. communis*. The divergent analysis revealed that the CTT and *R. communis* shared a common ancestor that diverged from the common ancestor of *H. brasiliensis*, *J. curcas*, and *M. esculenta* 36.7 million years ago (MYA) (Fig. 2c). The data also suggested that the CTT and *R. communis* diverged at 32.8 MYA.

A further investigation of gene cluster analysis revealed numerous genes showing expansion or contraction in the 12 genomes. For the CTT, there were 5287 and 4938 expanded and contracted orthogroups (Fig. 2c), respectively. Of the 5287 expanded orthogroups, a total of 53 orthogroups, including 762 genes, reached the significantly expanded level ($P < 0.05$) (Supplementary Data 10). Functional analysis of these 762 genes revealed that the top 5 enriched biological processes were macromolecule metabolic process, organic cyclic compound metabolic process, cellular aromatic compound metabolic process, and regulation of cellular process (Fig. 2d). Most of these biological processes are related to the biosynthesis of complex secondary metabolites that would enable plants to obtain higher resistance to various stresses. Additionally, there were numerous genes annotated as *resistance* (R) genes in 53 orthogroups (Supplementary Data 10). Therefore, the functions of these expanded genes in the CTT genome suggest that these genes may enable the CTT to obtain high stress or disease resistance. To investigate what types of duplications contribute to the formation of these significantly expanded genes, gene duplication types for the whole CTT monoploid genome were analyzed: in total, 3527, 4356, 1005, 1802, and 21,889 for a singleton, dispersed, proximal, tandem, and WGD (Supplementary Data 11), respectively; for the 762 significantly expanded genes, there were 2, 100, 217, 141, and 302, respectively.

Obviously, the proximal and tandem duplications in the 762 genes were significantly increased ($P <0.05$, Chi-square test). Usually, proximal and tandem duplications are considered the consequence of tandem duplication of large fragments or single genes[18,19]. Thus, these analyses suggest that the major driver of expanded genes is tandem duplications in the CTT genome.

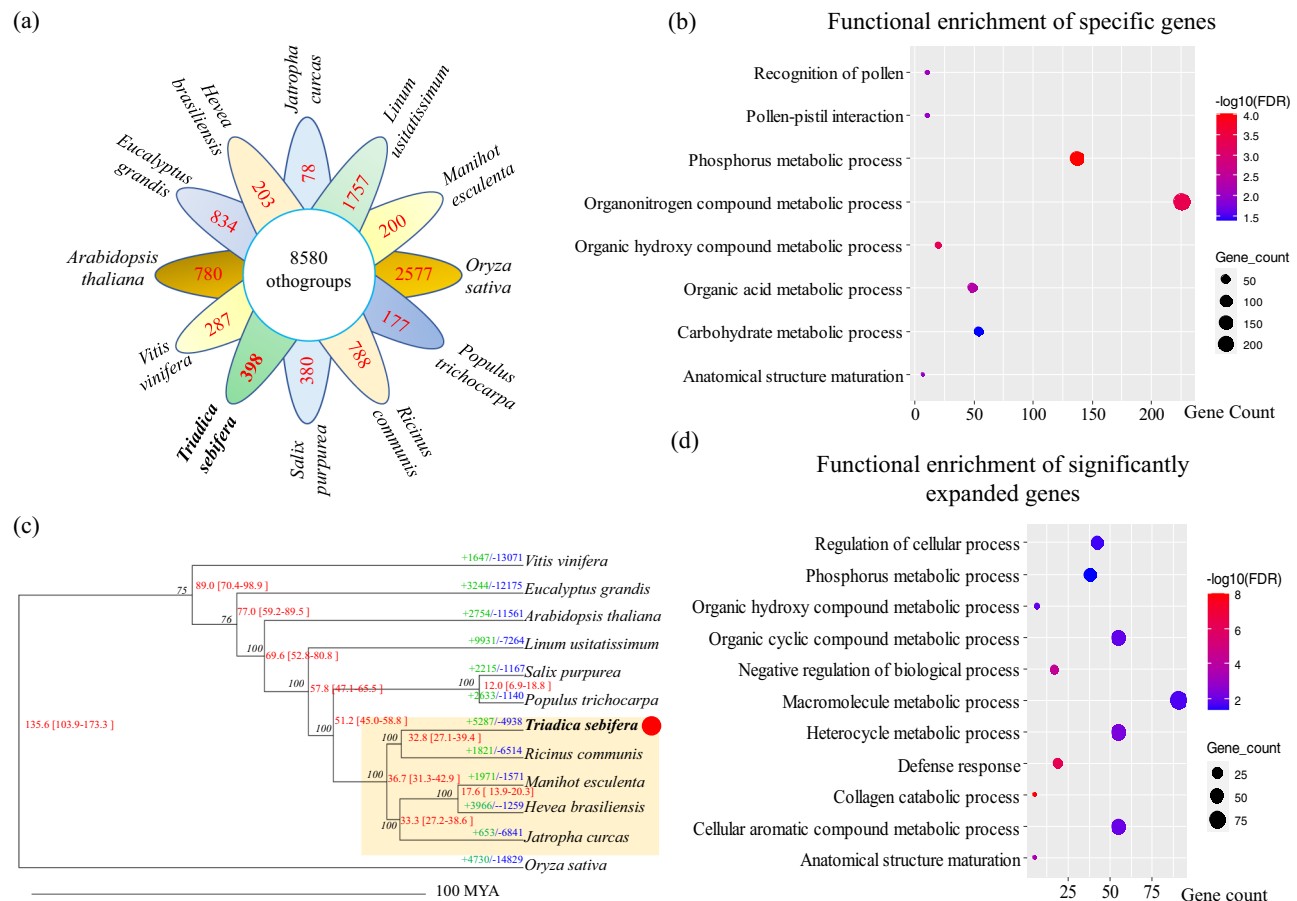

**Fig. 2 Genome expansion and phylogenetic analyses for the Chinese tallow tree (CTT). a** Orthogroups in the 12 plants, namely, CTT, *Arabidopsis thaliana*, *Eucalyptus grandis*, *Hevea brasiliensis*, *Jatropha curcas*, *Linum usitatissimum*, *Manihot esculenta*, *Oryza sativa*, *Populus trichocarpa*, *Ricinus communis*, *Salix purpurea*, and *Vitis vinifera*. **b** Functional enrichment of specific genes in the CTT genome. FDR represents the false discovery rate. **c** Phylogenetic tree for 12 plants listed in **a**. Single-copy genes were used to construct this tree with the PROTGAMMAAUTO model in RAxML. "+" and "−" indicates gene expansion and contraction, respectively. The green and blue numbers indicate numbers of othogroups that expand or contract, respectively. The red numbers indicate divergent time. For example, 89.0 [70.4–98.9] indicates the average divergent time is 89.0 million years ago (MYA), and its 95% confidence interval is 70.4 to 98.9 MYA. Plants with yellow backgrounds belong to Euphorbiaceae family. **d** Functional enrichment of significantly expanded genes in the CTT genome.

In the above analyses, the CTT and *R. communis* were predicted to share a common ancestor. To compare the two genomes, the monoploid genome of the CTT was first aligned with the genome of *R. communis*. Clearly, most collinear blocks in the *R. communis* genome matched 2 counterparts (Fig. 3a), suggesting that a genome duplication occurred after the divergence between the CTT and *R. communis*. Moreover, a self-comparison of the monoploid genome of the CTT also suggested that there is another collinear block for most chromosomes (Fig. 3b). In the K-mer analysis and genome assembly, we predicted that the genome of the CTT is tetraploid. To prove this prediction, the 63,207 genes predicted for the diploid CTT genome were also used in this analysis. The 4DTv values of the other four plants in Euphorbiaceae, namely, *H. brasiliensis*, *J. curcas*, *M. esculenta*, and *R. communis*, and the monoploid and diploid genome sequences CTT were calculated. The distribution of these 4DTv values revealed that all the plants, excluding the monoploid genome of the CTT, shared a common peak at ~0.37 (Peak 3) (Fig. 3c). This value is equal to a previous report for *V. vinifera*[4]; thus, the value suggested that there would be duplication in the common ancestor of these plants and that this duplication occurred outside the Euphorbiaceae family. The unusual peak of the monoploid genome of the CTT at 0.468 may be attributed to confusion of paralogs within its genome during the analysis. Only

the genomes of *R. communis* and *J. curcas* showed one 4DTv peak, while the other four genomes showed at least two peaks. The diploid and monoploid genomes of the CTT showed a similar value of Peak 2 (ca. 0.05). Additionally, the other two plants, *H. brasiliensis* and *M. esculenta*, showed values of 0.081 and 0.123 at Peak 2, respectively. These values of Peak 2 suggest that the genomes of all 3 plants were duplicated after they diverged from their common ancestor of *R. communis* or *J. curcas*. Moreover, the diploid genome of the CTT also shows the third unique peak (Peak 1, 0.002), and the value is much less than the value of the other two peaks. This result suggests that there was a very recent duplication of the CTT genome. According to the above analyses, we predicted an evolutionary model for the Euphorbiaceae family (Fig. 3d). In this model, there was a common duplication outside this family, one duplication occurring in *H. brasiliensis* and *M. esculenta* and two additional duplications in the CTT genome. The two duplications in the CTT genome highly support our prediction of the tetraploidy of this plant.

**Dissection of biosynthesis pathways of fatty acids (FAs).** The seed of CTT produces stillingia oil inside the kernel and tallowy fat in the white coat outside the kernel (Fig. 4a). After pollination,

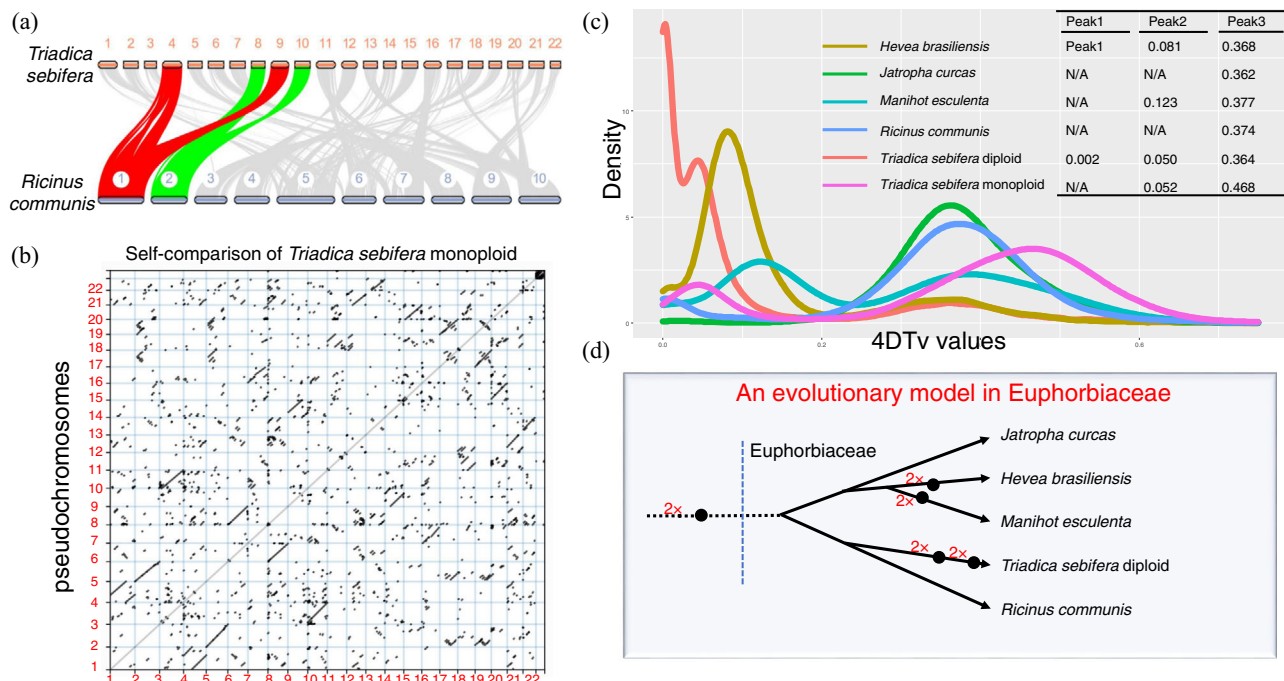

**Fig. 3 Genome duplication analysis for the Chinese tallow tree (CTT). a** Synteny comparison of the CTT monoploid genome with *Ricinus communis* genome. Bars indicate chromosomes or scaffolds, while lines indicate synteny blocks between the two genomes. Synteny blocks for chromosomes 1 and 2 in the *R. communis* genome are highlighted with red lines, which indicates that one locus matches two counterparts in the CTT genome. **b** Self-comparison of the CTT monoploid genome. Both the x- and y- axes indicate the scaffold of the CTT monoploid genome. **c** Frequency plot of the fourfold synonymous (degenerative) third-codon transversion (4DTv) values for six genomes, namely, *Hevea brasiliensis*, *Jatropha curcas*, *Manihot esculenta*, *Ricinus communis*, CTT diploid, and CTT monoploid. **d** An evolutionary model predicted for some plants in the Euphorbiaceae family. A black dot indicates that a whole genome duplication occurred. The line length roughly indicates divergent time.

ca. 4 months is required for the development of a mature seed for the CTT (Fig. 4a). The FA compositions of stillingia oil and tallowy fat are totally different. There is ca. 60% palmitinic acid (C16:0) and 30% oleic acid (C18:1) in tallowy fat and 30% linoleic acid (C18:2) and 50% linolenic acid (C18:3) in stillingia oil[9,10]. Thus, there is a high accumulation of saturated and unsaturated FA in the coat and kernel, respectively.

To dissect the biosynthesis pathways of FA in the seeds of the CTT, a homology search of genes involved in this pathway was conducted by using the information released in the Acyl Lipids database (http://aralip.plantbiology.msu.edu/pathways/pathways). Seed samples of four different developmental stages, namely, 1, 3, 5, and 16 weeks after pollination (WAP), were also collected (Fig. 4a). To compare the FA-related genes with other plants, a homology search of genes involved in the FA pathway was conducted for *R. communis*, *J. curcas*, *H. brasiliensis*, and *M. esculenta* (Fig. 4b). Clearly, the CTT monoploid genome harbors more genes for most FA-related enzymes, especially HAD (hydroxyacyl-ACP dehydrase), LACS (long-chain fatty acyl-CoA synthetase), FAD2 (fatty acid desaturase 2), and FAD3 (fatty acid desaturase 3) (Fig. 4b). Specifically, the much higher gene numbers found for the CTT than other plants for FAD3, which are the determinant genes for unsaturated FA biosynthesis, highly agree with the high accumulation of linolenic acid in stillingia oil. Moreover, all three *FAD3* genes were produced by WGD (Supplementary Data 12). The expression profiles of all these FA biosynthesis genes were also investigated in leaves and the four developmental stages of seeds. Except for genes encoding LACS, which is responsible for long-chain FA biosynthesis, most of the genes that can be detected to have expression values (RPKM >1 in at least one biological sample) showed higher expression levels in seeds than leaves (Fig. 4b and Supplementary Data 12).

Interestingly, a total of six genes encoding LACS showed much higher expression in leaves than in seeds. Moreover, two *FAD2* and two *FAD3* genes showed much higher expression in developmental stages 3 and 4, both of which would be the major phases for the accumulation of unsaturated FAs according to the development of seed (Fig. 4b and Supplementary Data 12). Therefore, more genes encoding *FAD3*, higher expression of *FAD2* and *FAD3* in seeds, and less expression of some genes encoding LACS that will convert oleic acid into longer-statured FAs may be responsible for the high accumulation of linoleic and linolenic acids in stillingia oil.

**Identification of candidate genes for the formation of red leaves.** The CTT is an important landscaping tree species and can show charming red leaves in autumn and winter (Fig. 1a). However, the leaves of the CTT before autumn are green (Fig. 5a). Additionally, the young leaves are sometimes light red. To investigate the mechanisms underlying the formation of red leaves, three types of plant materials, namely, green, light red, and red leaves, were collected (Fig. 5a). Anthocyanins in these samples were extracted and then examined by UHPLC-MS/MS. The products were compared with numerous known metabolites of anthocyanins. In positive ionization mode, the product showed parent and daughter ion peaks at m/z 449.11 and 287.06, respectively (Fig. 5b). This pattern is identical to the spectrum of Cyanidin 3-O-glucoside (Cy3-g) in a previous report[20]. Therefore, the major composition of anthocyanin in leaves of the CTT is Cy3-g. The content of this metabolite in green, light red, and red leaves was $0.12 \pm 0.0$, $9.30 \pm 0.2$, and $139.26 \pm 1.35$ ng/g fresh weight (FW), respectively (Fig. 5c).

Because the anthocyanin biosynthesis pathway has been well studied in *V. vinifera* and *A. thaliana*, structural genes encoding enzymes in this pathway were used as queries to search the

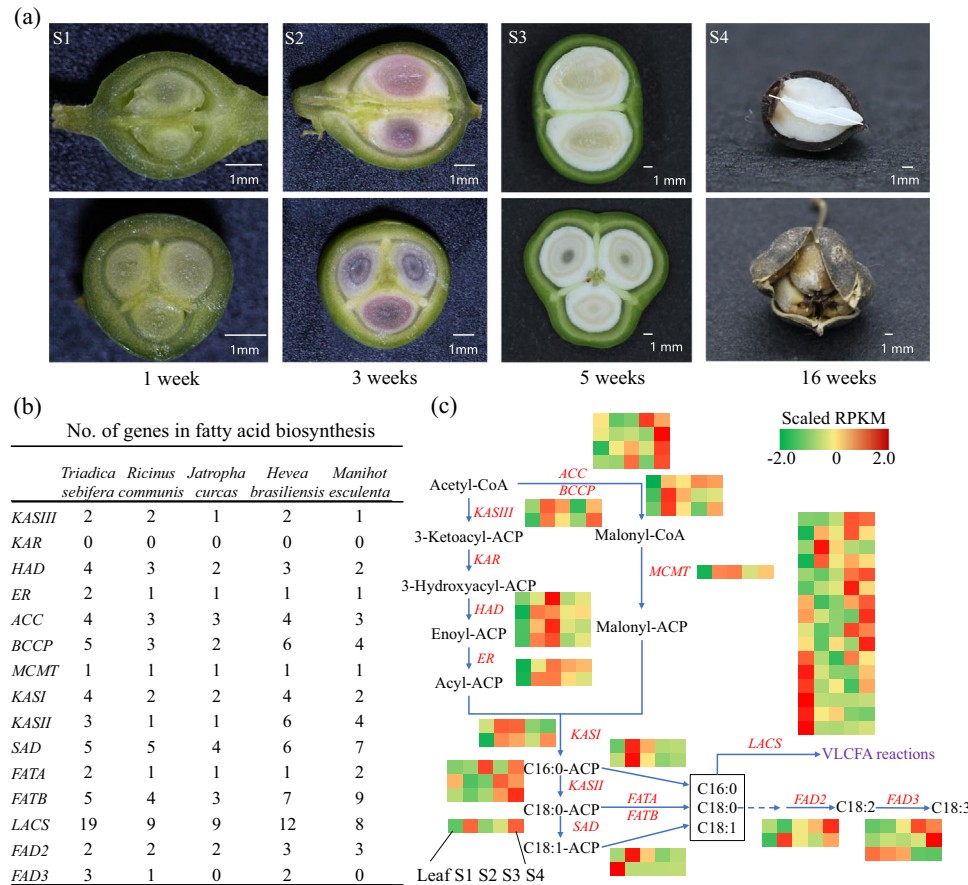

**Fig. 4 Analysis of fatty acid (FA) biosynthesis genes in the Chinese tallow tree (CTT) genome. a** The development of CTT seeds. S1 to S4 indicate 1, 3, 5, and 16 weeks after pollination (WAP), respectively. The upper panels indicate transverse cross sections of CTT seeds in different developmental stages, while the bottom panels indicate the longitudinal sections of CTT seeds. **b** Numbers of genes involved in fatty acid biosynthesis for five plants in the Euphorbiaceae family. The monoploid genome sequence was used to perform this analysis. **c** Gene expression patterns of FA biosynthesis genes in leaves and S1 to S4 developmental stages. The five boxes with different colors indicate these five samples. Scaled reads per kilobase per million mapped reads (RPKM) values were used for the heat plot. KASIII β-ketoacyl-acyl carrier protein synthase-III, KAR 3-ketoacyl-acyl carrier protein (ACP) reductase, HAD 3-hydroxyacyl-ACP dehydratase, ER enoyl-acyl carrier protein reductase, ACC acetyl- CoA carboxylase, BCCP biotin carboxyl carrier protein, MCMT malonyl-CoA ACP transferase, KASI β-ketoacyl-acyl carrier protein synthase-I, KASII β-ketoacyl-acyl carrier protein synthase-II, SAD stearoyl-ACP desaturase, FATA fatty acyl-ACP thioesterase A, FATB fatty acyl-ACP thioesterase B, LACS long-chain acyl-coenzyme A (CoA) synthetase, FAD2 fatty acid desaturase 2, FAD3 fatty acid desaturase 3.

genome of the CTT. A total of 23 genes encoding the seven enzymes, chalcone synthase (CHS), chalcone isomerase (CHI), flavanone 3-hydroxylase (F3H), anthocyanidin synthase (ANS), dihydroflavonol 4-reductase (DFR) and leucoanthocyanidin dioxygenase (LDOX) and uridine diphosphate-glucose:flavonoid 3-O-glucosyltransferase (UFGT) and O-methyl transferase (OMT), shown in Fig. 5d were identified (Supplementary Data 13). The expression of these 23 genes in green and red leaves was also calculated and compared by using RNA-Seq data. Interestingly, all nine differentially expressed genes (DEGs) showed higher expression in red leaves than in green leaves (Fig. 5d). Moreover, at least one DEG was identified for six enzymes, CHS, CHI, F3H, ANS, DFR, LDOX, and UFGT, suggesting that the higher expression of these genes in the red leaves would be attributed to the formation of red color for the leaves of the CTT. Genes of the *MYB-bHLH-WD40* (*MBW*) complex, *ELONGATED HYPOCOTYL5* (*HY5*), and *CONSTITU-TIVE PHOTOMORPHOGENIC1* (*COP1*) were reported to regulate structural genes in anthocyanin biosynthesis in a number of studies[21] (summarized in Fig. 5d). Thus, DEGs of these genes in the CTT were also identified, except *WD40* genes, due to a

large number of genes in this gene family in the genome. In total, seven *MYB* genes and one *bHLH* gene showed significantly higher expression in red leaves than in green leaves (False discovery rate <0.01, fold change >2.0). Therefore, these 8 transcription factors (TFs) could be the determinant genes for the formation of red leaves for the CTT in autumn and winter.

## Discussion

The CTT can grow well in a number of stress conditions in China and is now regarded as an invasive plant in southeastern USA[13]. In this study, we were able to provide the chromosome-scale genome sequence for the CTT. According to the analysis of its genome, the fact that the CTT is a tetraploid would be the most interesting finding. Usually, polyploid plants have more growth vigor than diploid plants[22,23]. Therefore, the tetraploid CTT genome would be the major diver of its high adaptability character. Additionally, we also found that some genes related to disease resistance, nutrition, energy utilization, and secondary metabolite biosynthesis are specific to the CTT or expand significantly in its genome (Fig. 2 and Tables S7, S8). These genes

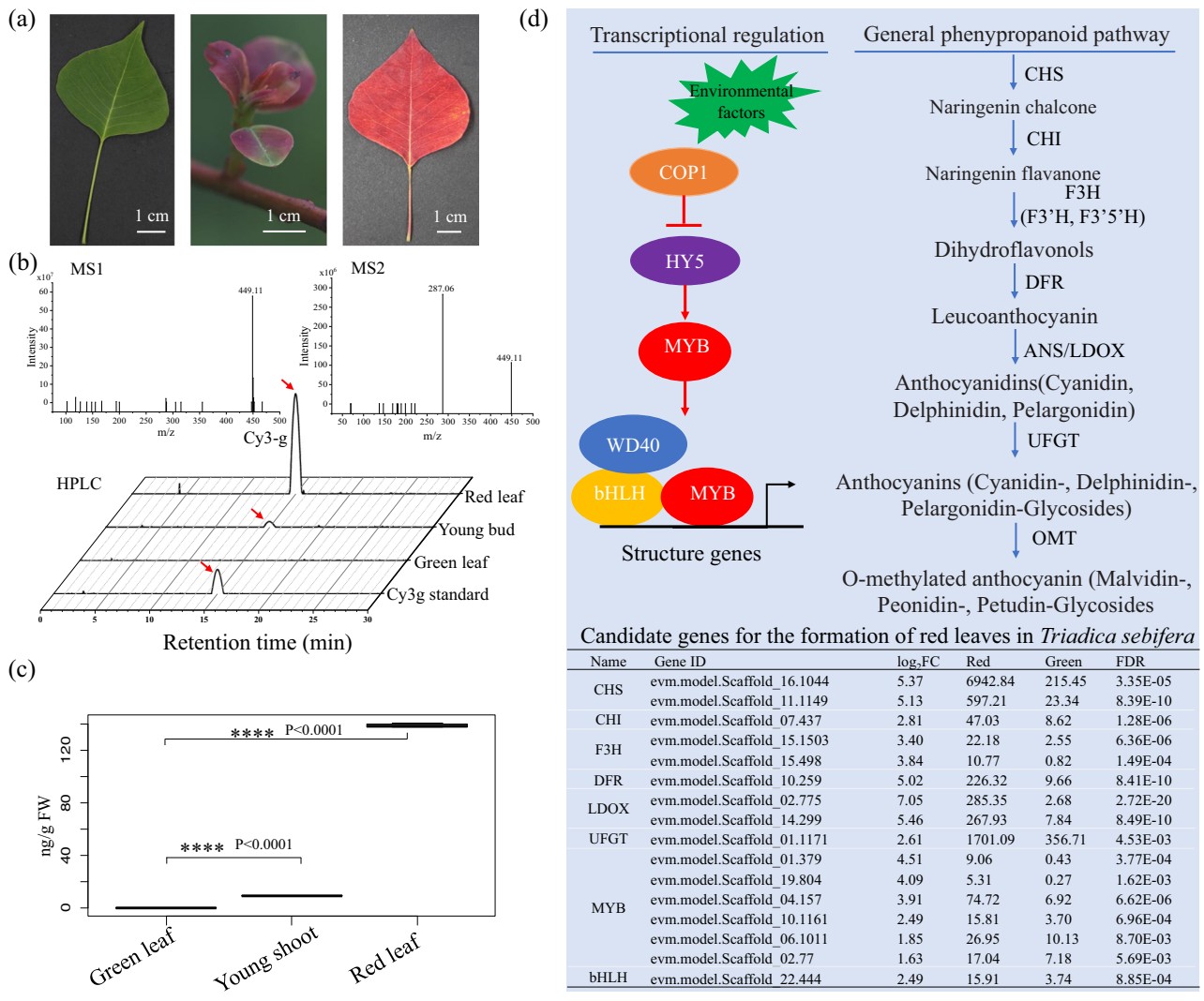

**Fig. 5 Analysis of anthocyanin biosynthesis genes in the Chinese tallow tree (CTT) genome. a** Three types of leaf samples with different colors were used for anthocyanin measurement and RNA-Seq analysis. **b** Determination of the major composition of anthocyanin in CTT leaves. Anthocyanins in red leaves were extracted and examined by using a Q exactive plus ultra-high-performance liquid chromatography instrument with a mass spectrometer (UHPLC/MS) (Thermo Fisher, USA) (upper panel), and the content of Cyanidin 3-O-glucoside (Cy3-g) was measured with high-performance liquid chromatography (HPLC) (Waters Alliance-e2695, Waters Corporation, USA) instrument (lower panel). **c** Contents of Cy3-g in three types of leaf samples. **d** Anthocyanin biosynthesis genes in the CTT genome. The upper panel indicates the regulation and structure genes for anthocyanin analysis. The bottom panel indicates differentially expressed genes (DEGs) between the red and green leaves for the regulation and structure genes of anthocyanin in the CTT. Columns in this table from left to right are enzyme name, gene ID, Log$_2$FC between red to green leaves, reads per kilobase per million mapped reads (RPKM) values for red leaves, RPKM for green leaves, false discovery rate (FDR) for the comparison of red to green RPKM. The full names of these enzymes are CHS chalcone synthase, CHI chalcone isomerase, F3H flavanone 3-hydroxylase, ANS anthocyanidin synthase, DFR dihydroflavonol 4-reductase, LDOX leucoanthocyanidin dioxygenase, UFGT uridine diphosphate-glucose:flavonoid 3-O-glucosyltransferase, OMT O-methyl transferase, and bHLH basic helix-loop-helix.

would also be another driver for the tree's high adaptability character.

In addition, the tetraploid genome enables CTT to get high adaptability, however, this character increases the difficulty in genome assembly. In this study, we were not able to obtain the chromosome-scale sequence for the CTT diploid genome, and only the purged chromosome-scale monoploid genome was created for subsequent analyses. Though we found the monoploid genome could represent most gene information in the diploid genome (Supplementary Data 8), it was still possible that some important sequence information was lost. Thus, a high-quality chromosome-scale diploid genome resource is still required in future studies.

According to the calculation of 4DTv values for five plants in the Euphorbiaceae family, all plants show peak values ca. 0.37,

suggesting that a common duplication occurs for all five plants. This peak at 0.37 was also detected in *V. vinifera* in a previous report. Because an ancient duplication is considered to occur in the *V. vinifera* genome[24], plants in Euphorbiaceae also harbor this ancient WGD. Within the Euphorbiaceae family, there are no additional 4DTv peaks for *J. curcas* and *R. communis*, which suggests that no additional WGD occurs in these two plants. Meanwhile, both *H. brasiliensis* and *M. esculenta* show the second peak (Peak 2, Fig. 3c); however, their peak values are not the same. Therefore, these data suggest that WGD occurs in these two plants at different times. The CTT showing small values at Peak 2 and Peak 3 may suggest that the two WGD events occurred recently (Fig. 3d).

The initial assembly for monoploid and tetraploid genomes CTT are 2.88 Gb and 787 Mb, respectively, while the genome

sizes for *J. curcas*, *R. communis*, *H. brasiliensis*, and *M. esculenta* are 264, 336, 1590, and 742 Mb[2–4,25], respectively. The genome sizes of these five plants in the Euphorbiaceae family seem not to match the WGD and polyploidization levels in their genome. However, after gaining insight into the size of LTRs in these genomes, it is easy to explain these inconsistencies. LTR takes up ca. 50 and 65% of the whole genomes of the CTT and *H. brasiliensis*, respectively (Table 1)[2], while the other three genomes show lower proportions of LTRs[2,4,25]. Thus, these data suggest that a burst of LTRs is the major contributor to genome size.

In seeds of the CTT, two types of oil, stillingia oil and tallowy fat, are accumulated. There is a very high content of unsaturated FAs (C18:2 and C18:3) in stillingia oil[8–10]. FAD2 and FAD3 are considered the two major enzymes for the biosynthesis of C18:2 and C18:3. In this study, we found two and three genes encoding FAD2 and FAD3 in the CTT monoploid genome (Fig. 4b), respectively. In the Euphorbiaceae family, there was no gene encoding FAD3 in the *J. curcas* and *M. esculenta* genome, while one and two genes in *R. communis* and *H. brasiliensis* genomes, respectively. This data further indicates that there are many more genes encoding FAD3 in tetraploid of the CTT (the number would be 12) than all other species in the Euphorbiaceae family. In further analysis, we found that the 3 *FAD3* genes were produced by WGD in the CTT monoploid genome (Supplementary Data 12). In our above analysis, two rounds of WGD occurred in the CTT monoploid, *M. esculenta* and *H. brasiliensis* genomes, and they shared the first round of WGD (Fig. 3c, d). However, there are only 0 and 2 *FAD3* genes in *M. esculenta* and *H. brasiliensis* genomes (Fig. 4b), respectively. The less *FAD3* genes in *M. esculenta* and *H. brasiliensis* suggested that *FAD3* was not duplicated or deleted in the second round of WGD. In other words, more *FAD3* genes in the CTT monoploid genome are mainly attributed to the second round of WGD.

In the dissection of the formation of attractive leaf color in the CTT, we first uncovered Cy3-g was the major composition of anthocyanins in red leaves (Fig. 5a–c). The identification of structural genes for anthocyanin biosynthesis revealed that genes encoding six enzymes were differently expressed in red and green CTT leaves. Especially one gene *evm.model.Scaffold_01.1171* encoding an anthocyanidin 3-O- glucosyltransferase showed ca. fivefold higher expression in red than green CTT leaves. The anthocyanidin 3-O-glucosyltransferase in Arabidopsis specifically glycosylates the 3-position of the flavonoid C-ring[26,27]. Because Cy3-g is the major component of anthocyanins in red leaves of CTT, our results are consistent with studies reported in Arabidopsis[26]. This indicates the expression of these genes encoding the 8 TFs in Fig. 5d are activated in the autumn and winter, which suggests these genes might play vital roles in formatting red leaves in autumn and winter. However, the mechanisms for the increased expression of these genes are not clear. In Arabidopsis, the gene encoding anthocyanidin 3-O-glucosyltransferase showed increased expression due to a change in light (increased light/dark ratio) under cold conditions[28]. This might also suggest that *evm.model.Scaffold_01.1171* would also respond to light change and cold conditions. The red color of CTT leaves is formed in the late autumn and early winter, thus, photoperiod, light intensity, and the environmental temperature are also changed in this growth condition. Therefore, the activated expression of genes related to anthocyanin biosynthesis in the CTT would also be attributed to the changed conditions of light and temperature.

In conclusion, a chromosome-scale genome was assembled for the CTT in this study, and it provided a valuable genomic resource for this plant. According to the comprehensive analyses of this valuable genome resource, we uncovered novel knowledge to understand the unique characteristics of this plant, such as

high adaptability, high content of unsaturated FAs in seeds, and attractive leaf color in autumn and winter. Moreover, we also found two recent WGDs of the CTT genome, and these duplications made it a tetraploid. These results will help us to better utilize this plant in the future.

## Methods

**Plant materials.** A CTT line was found in the field of Luotian County, Hubei Province, China (N 31.05°, E 115. 66°, H 387.7 m) (Fig. 1a). Seeds of this CTT line were harvested at 1, 3, 5, and 16 WAP were used for RNA isolation in 2020. A total of 5–10 seeds were mixed as one biological sample and three biological samples were collected at each time point. Meantime, the red and green leaves were also collected from the same CTT line. The red and green leaves were collected at different time points. Similarly, three biological samples were collected for green and red leaves, respectively. These samples were then used for anthocyanin determination and RNA isolation. This CTT line was then propagated in a woody plant medium (WPM) by using young shoots as the initial explants. Tissue culture plants were then grown in a greenhouse for further experiments.

**Genomic sequencing.** DNA was isolated from the young CTT plants that propagated from the CTT line found in Luotian County by using the modified cetyltrimethylammonium bromide (CTAB) method[29]. RNA was isolated from seeds and green and red leaves of the CTT by using an RNeasy Plant Mini Kit according to the manufacturer's instructions (DP432, TIANGEN Biotech (Beijing) Co., Ltd., Beijing, China). DNA and RNA that met the required quality were sent for sequencing. For pair-end (PE) read genomic sequencing and RNA sequencing (RNA-Seq), 150-bp PE libraries were constructed, and these libraries were sequenced by using the MGISEQ-2000 platform (BGI, Shenzhen, China).

Isoform sequencing (Iso-Seq) of mRNA and long high-fidelity (HiFi) sequencing of genomic DNA were conducted by the Single-molecule real-time (SMRT) Pacific Biosciences (PacBio) platform. The Hi-C library was also constructed using fresh leaves of the young tissue culture plants of the CTT, and it was sequenced on the Illumina HiSeq platform NovaSeq 6000 (Illumina, San Diego, CA). All library construction, sequencing, and raw read filtering was conducted according to the manufacturer's instructions.

**K-mer estimation and flow cytometry detection.** The 150-bp PE DNA reads of the CTT were filtered by using Trimmomatic software with default parameter[30]. All PE reads were also filtered by using this software in further analyses. The clean PE reads were then used for K-mer analysis. The distribution of the 17-bp K-mer was calculated with "kmerfreq" implemented in GCE software with parameter setting as "-k 17"[31]. Genome size and repeat content were estimated with "gce" program with parameter setting as "-H 1". Young leaves of the CTT and the poplar line "NL895" were harvested for flow cytometry detection using a cell analyzer (BD LSRFortessa, BD Biosciences, New Jersey, USA). This popular line was used in our recent studies. The preparation of samples and fluorescence detection were performed according to the manufacturer's instructions. The monoploid genome size of poplar "NL895" was set as 456 Mb[17], and it was used as the control sample to estimate the genome size of the CTT.

**Genome assembly.** Two software programs, Canu (Canu v2.1.1) and Hifiasm (HifiAsm-0.16.1)[32,33] were used to conduct genome assembly by using HiFi reads. The produced contigs of Canu were purged by using purge_dups (v1.0.1) (https://github.com/dfguan/purge_dups). The raw contigs produced by Canu and final purged contigs produced by purge_dups were used as the tetraploid and monoploid genomes of the CTT for further studies, respectively. For the HifiAsm assembly, different parameters were used, and the final parameter was "-l 3" to obtain the diploid genome of the CTT for further studies. All three types of assemblies were assessed by using Benchmarking Universal Single-Copy Orthologs (BUSCO) analysis (The lineage dataset: embryophyta_odb10, Creation date: 2020-09-10, number of species: 50, number of BUSCOs: 1614)[34].

The clean Hi-C reads were filtered to produce valid reads by using the software of HiCUP (Version 0.8.2) with default parameters[35]. The clean and valid Hi-C reads were then mapped onto the monoploid genome of the CTT using Juicer software with default parameters[36]. The candidate chromosomes/scaffolds were generated by 3d-DNA pipelines with a "haploid" mode[37]. The resulting chromosomes/scaffolds were manually corrected via Juicebox Assembly Tools[38].

**Repeat sequence identification, gene prediction, and functional annotation.** Transposable elements (TEs) were identified by using a combination of homology-based and de novo approaches. The software programs Repeat-ProteinMask, RepeatModeler 2.0, LTR Finder v. 1.0.6, and RepeatMasker v.4.0.5 were used to perform this analysis[39–42].

Three strategies were employed for gene prediction, namely, ab initio prediction, homology-based prediction, and transcriptome-based prediction. The repeat-masked chromosome-scale genome of the CTT monoploid and the diploid contigs were used for gene prediction. First, CTT genome sequences were aligned

with protein sequences of five plants, namely, *A. thaliana*, *V. vinifera*, *O. sativa*, *P. trichocarpa*, and *J. regia*, and gene structures were predicted with the Exonerate pipelines version exonerate-1.0.0 with default parameters[43]. Second, RNA-Seq and Iso-Seq data were used to predict gene structures with the Program to Assemble Spliced Alignments (PASA) version PASApipeline.v2.4.1[44]. Third, ab initio prediction was performed with Augustus (https://github.com/Gaius-Augustus/Augustus) and GlimmerHMM (version GlimmerHMM-3.0.4)[45,46]. Fourth, all gene models were integrated with EVidenceModeler (EVM, version 1.1.1) to generate a consensus set[47]. Finally, the integrated gene models were updated with PASA pipelines[44].

The protein sequences of all gene models were aligned with the Swiss-Prot, TrEMBL, TAIR, and Nr databases with the diamond blastp program[48]. All these databases were downloaded on 2021-09-01. The descriptions or Gene Ontology (GO) terms were assigned with an automated assignment of human-readable descriptions (AHRD) pipelines (https://github.com/groupschoof/AHRD). Functional enrichment analyses were performed with TBtools pipelines[49]. All above procedures and parameters were set according to the manufacturer's instructions.

**Divergence time estimation, gene family expansions and contractions, synteny analysis**. Gene families among 12 plants were identified with OrthoFinder (version_2.5.4)[50]. Briefly, the longest protein sequence of each gene model of the 12 plants was extracted, and an all-by-all blastp was conducted. The orthogroups were then identified with default parameters implemented in OrthoFinder software. The phylogenetic trees of these 12 plants were constructed with single-copy genes by using the software RAxML (the model parameter was set as "PROTGAMMAAUTO")[51]. Expansion and extraction of gene families were analyzed using Computational Analysis of gene Family Evolution (CAFE, version 3.1)[52]. The divergence times for these 12 plants were estimated using the MCMCTree program[53]. The time tree was calibrated by using the known divergence time for some pairs in the 12 plants (http://www.timetree.org/). The synteny between the CTT and castor bean or within the CTT monoploid genome was identified by JCVI pipelines (https://github.com/tanghaibao/jcvi). The fourfold synonymous (degenerative) third-codon transversion (4DTv) values for each gene pair were calculated using KaKs_Caculator[54]. Gene duplication types were identified with MCScanX software[19].

All above procedures and parameters were set according to the manufacturer's instructions. All parameters except those mentioned above were set as default.

**Gene expression analysis**. Clean reads of RNA-Seq data were mapped onto the monoploid genome of the CTT using hisat2, and read counts for each gene were calculated using featureCounts[55,56]. Differentially expressed genes (DEGs) were called using edgeR[57]. The parameters of all these analyses were set as default. Differently expressed genes were set as false discovery rate (FDR) <0.01 and fold change >2.0 in a comparison pair.

**Determination of anthocyanin**. A total of 0.2 g leaves were harvested for one biological sample. Anthocyanin extraction was performed according to the standard protocol[58]. The extracted anthocyanins were analyzed using a Q Exactive Plus ultra-high-performance liquid chromatography instrument with a mass spectrometer (UHPLC/MS) (Thermo Fisher, USA) and high-performance liquid chromatography (HPLC) (Waters alliance-e2695, Waters Corporation, USA) instrument. After the extracted anthocyanins were taken from −80 °C and centrifuged at 12,000 rpm for 30 min, 150 μL supernatant was taken in the internal cannula of the sample vial. UHPLC/MS and HPLC analysis were performed using previously reported conditions[59].

**Identification of genes in pathways of fatty acid (FA), anthocyanin, and resistance**. Genes in the FA biosynthesis pathway in the genome of *A. thaliana* were retrieved from the Acyl Lipids database (http://aralip.plantbiology.msu.edu/pathways/pathways). Genes in the anthocyanin biosynthesis pathway in the genome of *A. thaliana* and *V. vinifera* were summarized from some previous reports[60,61]. Gene sequences in the target genome were then used to search all genes of *A. thaliana* or *V. vinifera*. The Pfam domain of all genes in these three plants was also identified by using the online software eggnog (http://eggnog-mapper.embl.de/)[62]. Three criteria were used to define FA- or anthocyanin-related genes in the CTT genome: (i) The target gene is the top hit of FA- or anthocyanin-related genes in the *A. thaliana* or *V. vinifera* genomes; (ii) the blast E-value is less than 10e$^{-10}$; and (iii) the candidate gene harbors the same Pfam domain as the queried gene for resistance (R) identification.

**Statistics and reproducibility**. Statistical analyses were conducted using the statistical computing programming language R (version 4.0.5). All statistical analyses were conducted with a significance level of α = 0.05 ($p \leq 0.05$). Sample sizes and replicates are demonstrated in the corresponding descriptions.

**Reporting summary**. Further information on research design is available in the Nature Research Reporting Summary linked to this article.

## Data availability

The whole genome sequence data, including PE short reads, HiFi reads, Hi-C interaction reads, transcriptome data, and genome files, have been deposited in the NCBI under accession number PRJNA813698.

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

## Acknowledgements

Financial support for this work was provided by the Fundamental Research Funds for the Central Universities (No. 2662022YLYJ007), Key Scientific and Technological Grant of Zhejiang for Breeding New Agricultural Varieties (2021C02070-7), and Special Support Funds of Zhejiang for Scientific Research Institutes (2021F1065-12).

## Author contributions

J.L., W.R., G.C., L.H., X.S., N.L., C.N., and Y.L. conducted the experiments. N.W., Y.L. and J.L. wrote and edited the manuscript. N.W. organized and supervised the whole project.

## Competing interests

The authors declare no competing interests.
