## [Peer Review File · Communications Biology]

Reviewers' comments:

Reviewer #1 (Remarks to the Author):

In their manuscript, Luo et al. reported a chromosome-scale genome of the Chinese tallow tree and described annotation, evolution, oil metabolism and anthocyanin formation genes. In my view, some issues need further consideration.

Description about the Chinese tallow tree, however, is not enough. This plant is also known as a traditional Chinese medicinal plant. I wonder why the authors limit use of this plant to energy oil obtained from the seeds and attractive leaf color. Description and discussion about these (and other aspects of this plant, if any) are required for the general interest.

In addition, the ms needs to be substantially rewritten, especially the discussion part. In the discussion part, I didn't find any information about biosynthesis of FAs in seeds and anthocyanin in leaves, which was main conclusions in abstract and introduction. I think the authors should clarify the purpose of this ms and discuss them. I found there are some sentences should be re-worded, and in placed the languages needs to be improved. I recommend that the authors have the ms edited by a native speaker of English.

L66-80: This paragraph and the following paragraphs described some physiological characteristics of the spurge family and the Chinese tallow tree. Most descriptions are not generally known, and relevant literature should be cited.

L76: I didn't find any literature on the renaming of the Chinese tallow tree. The authors should attach relevant literatures here and clarify which name was used in this ms. The authors used CTT as the name of the research object, I don't think it is appropriate. Using the unique latin name would be better.

L92: Please clarify "...in these places".

L94: Please clarify "...the above analyses".

L107-127: It is inappropriate and illogical for the authors to display all conclusions here, just before the result part. This paragraph could only raise the scientific questions.

L183-186: Rich LTR in plants seem to be common, not only the Euphorbiaceae. Further, It is inappropriate to judge burst patterns only by the percentage.

L193: Please provide a venn diagram to show the annotated results more clearly. What did "Out of the 40654 proteins" mean? In line 189, you said 32579 genes encoding 43536 proteins.

L197: What are the karyotypes of these selected species? Why not use two sets of haplotypes for family expansion and contraction

L202: How to defined the CTT-specific genes? Please clarify.

L253-254: How to understand the sentence "may be attributed to a confusion of papralogs within its genome during the analysis".

L320: How many DEGs?

L321: What do you mean "the first 6 enzymes"? Further, why only one of "the first 6 enzymes" related gene showed differential expression and then you said, most of the structural genes in anthocyanin biosynthesis is activited. Does the DEG encode a rate-limiting enzyme? In addition, what did "NA" mean in Table S11? The title of Table S11 should be changed, because it contains results of differential expression analysis.

L323-324: "Thus, regulation of these... " The logic is misleded. No valid explanation has been provided.

L328-329: what do you mean "due to large number of these genes in the genome"? Please clarify.

L330: P-value or P-adjust?

L376: In Methods part, the authors should provide the version and citation of the programs and softwares. Detailed methods or pipeline are required, such as genome size estimation, the model of sequencing platform.

L393: Which model of the sequencing platform? Novaseq6000 or others?

L412: BUSCO, which lib was used?

L414: Did the clean hic reads are valid reads? How to generate the valid reads? Please provided more detailed information.

Fig1f, the authors should provide x-axis and y-axis information. Fig1g, complete the legend.

Fig2c, Required detailed explanation. For example, the means of words in red, green and blue. Fig2b,d, the authors should mark the p-adjust values.

Fig4c, the authors should provide group information in the heatmap x-axis.

Fig. 5d, bHLH or BHLH?

Reviewer #2 (Remarks to the Author):

This is an interesting study, and I look forward to seeing the revision. However, there are some significant issues with the methods and analyses as noted in the marked up manuscript. The major one is the likelihood that the true haploid genome of the tetraploid (44 chromosomes) was collapsed into a combined consensus (ancestral??) genome with only 22 chromosomes. Thus, the analyses are not accurate. The genome annotation and subsequent clustering and phylogenetic analysis would also benefit from a more euphorbia-centric approach. As it is, the tree that was generated lacks statistical significance. Also, there are significant gaps in the methods that need to be addressed. On a personal note, I am also curious where leafy spurge fits into the comparisons (<https://www.cambridge.org/core/journals/weed-science/article/abs/gene-space-and-transcriptome-assemblies-of-leafy-spurge-euphorbia-esula-identify-promoter-sequences-repetitive-elements-highquality-markers-and-a-fulllength-chloroplast-genome/95C85C6B07094B6FA308C00608FDB816>) - particularly since it is also an invasive species in the US.

Reviewer #3 (Remarks to the Author):

Manuscript COMMSBIO-22-0922, titled "The chromosome-scale genome sequence of the Chinese tallow tree provides insight into its polyploidization, high adaptability and biosynthesis of its high content of unsaturated fatty acids in seeds and anthocyanin in leaves" explores the genome of the Chinese tallow tree, describes its importance in Chinese culture - as a source of bioenergy and bioproducts, as an ornamental, and as a highly adaptable species. Based on the data presented, the authors conclude that the Chinese tallow tree, a member of the Euphorbiaceae family, is a tetraploid likely resulting from whole-genome duplications, which increased genes for resistance, nutrition and energy utilization - thus, contributing to its invasive nature. The authors also include data to characterize genes and potential pathways associated with seed and leaf fatty acids and leaf anthocyanins.

Overall, the manuscript is well written but the results section, particularly regarding the tetraploid nature, is a little wordy. Perhaps the manuscript could be more focused on the aspects of fatty

acids and anthocyanins and less on the potential mechanisms of invasiveness. Because the genome sequence reported for Chinese tallow tree was from its native environment and not compared with the genome of its invasive counterpart in north America, it is pure speculation to suggest that the different categories of genes identified, compared with other species, is responsible for its invasiveness. Indeed, on Lines 339 – 342, the authors do acknowledge that the genome data presented is from Chinese tallow tree collected in China and not the invasive species of the U.S.A. Many other factors could be equally responsible for the invasive nature of Chinese tallow tree in north America. For example, *Euphorbia esula* (leafy spurge) is a perennial Euphorbiaceae species native to Eurasia that was accidentally introduced into north America, where it has become invasive. Numerous factors including an abundance of underground adventitious buds and root buds that go through well-defined phases of dormancy, and the lack of predatory pest common to its native range have been well documented as leading causes of invasiveness. The introduction of leafy spurge biocontrol agents from its native range, such as flea beetles, has helped to keep this invasive species in check – particularly in rangeland environments of north America. Thus, lines 342 – 347 “...the tetraploid CTT genome would be the major driver for its high adaptability and invasive character. Additionally, we also found that some genes related to disease resistance, nutrition and energy utilization, and secondary metabolite biosynthesis are specific for the CTT or expand significantly in its genome (Fig. 2, Tables S7 and S8). These genes would also be another driver for the tree’s high adaptability and invasive character.”, are again speculative in nature – as are the inclusion of Lines 102 – 106 suggesting the underlying mechanisms for invasiveness are due to “higher flavonoid concentrations in the roots and lower concentrations of tannins compared with native plant populations”. Certainly, the authors understand that Chinese tallow tree can become invasive; however, the data presented in this manuscript is too definitive on the mechanism(s) responsible. Without sequencing the genome of Chinese tallow tree from an invasive environment, or at least conducting an RNAseq study to determine if the genes in question from this study are overrepresented in invasive environments, the authors might consider reducing the manuscript’s emphasis on invasiveness and keeping it more focused on fatty acids and anthocyanins.

Minor comments:

Line 67 – saying that “Numerous plants in this family can be used as bioenergy.” And then giving examples of products that are not associated with bioenergy (rubber and castor oil) is odd. Perhaps revise to “Numerous plants in this family can be used for generating bioenergy and bioproducts. For example...”

Line 293 – 294: “the RPKM>1 in at least one biological sample” this seems like a low level of stringency. An RPKM>2 across all biological replicates of a single treatment or time point for seed or leaves would be more appropriate.

Line 298: perhaps the authors could include a reference for “...in a previous report...”

Reviewer #1 (Remarks to the Author):

In their manuscript, Luo et al. reported a chromosome-scale genome of the Chinese tallow tree and described annotation, evolution, oil metabolism and anthocyanin formation genes. In my view, some issue need further consideration.

Description about the Chinese tallow tree, however, is not enough. This plant is also known as a traditional Chinese medicinal plant. I wonder why the authors limit use of this plant to energy oil obtained from the seeds and attractive leaf color. Description and discussion about these (and other aspects of this plant, if any) are required for the general interest.

Response: First of all, thank you very much for your very constructive comments and suggestions to our manuscript. These comments and suggestions help us to improve the manuscript thoroughly and greatly. Concerning to the use of the Chinese tallow tree, generally, it is one of the most important trees that produce oil in China in the long history. In recent years, this tree is also used as an ornamental tree benefit from its leaf color. Additionally, the Chinese tallow tree is widely distributed in China due to its high adaptability to varied growth conditions. We already described all these important points of this tree in the introduction section. To address your concern, we also described some other use (one type of herbal medicine) of this tree, see line 99-103.

In addition, the ms needs to be substantially rewritten, especially the discussion part. In the discussion part, I didn't find any information about biosynthesis of FAs in seeds and anthocyanin in leaves, which was main conclusions in abstract and introduction. I think the authors should clarify the purpose of this ms and discuss them. I found there are some sentences should be re-worded, and in placed the languages needs to be improved. I recommend that the authors have the ms edited by a native speaker of English.

Response: We added two subsections for FAs and anthocyanins in the discussion part. The manuscript was also sent to American Journal Experts who provides editing service for proof reading and editing.

L66-80: This paragraph and the following paragraphs described some physiological characteristics of the spurge family and the Chinese tallow tree. Most descriptions are not generally known, and relevant literature should be cited.

Response: Because very few studies focusing on the Chinese tallow tree were performed in past decades, thus, there were not enough reports for this tree. We have collected all relative literatures for the Chinese tallow tree as many as we could and all these literatures were cited in this manuscript. We provided citations for most descriptions and the rest are the general knowledge for us to understand this tree. To address your concern, we also added some literatures as many as we could.

L76: I didn't find any literature on the renaming of the Chinese tallow tree. The authors should attach relevant literatures here and clarify which name was used in this ms. The authors used CTT as the name of the research object, I don't think it is appropriate. Using the unique latin name would be better.

Response: In a number of previous studies, the latin name of the Chinese tallow tree was *Sapium sebiferum*. In the last version of Flora of China, the latin name of the Chinese tallow tree was *Sapium sebiferum*. In the latest version of Flora of China, its latin name is *Triadica sebifera* (you can search it at the online version of Flora of China,

<http://www.iplant.cn>, Chinese version). The description can also be found on Flora of North America (http://floranorthamerica.org/Triadica_sebifera). To make it clear, we changed the website of Flora of China to Flora of North America in this version. Because the full name of the Chinese tallow tree is too long and it was used many times in the whole manuscript, so, we used its abbreviation CTT in our manuscript.

L92: Please clarify "...in these places".

Response: revised.

L94: Please clarify "...the above analyses".

Response: We revised this description.

L107-127: It is inappropriate and illogical for the authors to display all conclusions here, just before the result part. This paragraph could only raise the scientific questions.

Response: We already revised this paragraph.

L183-186: Rich LTR in plants seem to be common, not only the Euphorbiaceae. Further, It is inappropriate to judge burst patterns only by the percentage.

Response: We already revised the description.

L193: Please provide a venn diagram to show the annotated results more clearly. What did "Out of the 40654 proteins" mean? In line 189, you said 32579 genes encoding 43536 proteins.

Response: We revised this description. The venn diagram is very complicated to see detailed information (Six terms make a lot of overlaps), so we provided a new supplementary table (table S6) for the annotated results.

L197: What are the karyotypes of these selected species? Why not use two sets of haplotypes for family expansion and contraction

Response: Because we could not get the chromosome-scale genome for two sets of genomes when using Hi-C data for anchoring contigs. We already described this issue in the second paragraph of section 2.1.

L202: How to defined the CTT-specific genes? Please clarify.

Response: We added "A total of 398 orthogroups were only found in the CTT genome.". This description would help to define the CTT-specific genes.

L253-254: How to understand the sentence "may be attributed to a confusion of papralogs within its genome during the analysis".

Response: Because the CTT monoploid is merged from all set of genomes, therefore, genes in the CTT monoploid assembly may be somewhat different from the real case in one CTT monoploid.

L320: How many DEGs?

Response: We added this information.

L321: What do you mean "the first 6 enzymes"? Further, why only one of "the first 6 enzymes" related gene showed differential expression and then you said, most of the structural genes in anthocyanin biosynthesis is activited. Does the DEG encode a rate-limiting enzyme? In addition, what did "NA" mean in Table S11? The title of Table S11 should be changed, because it contains results of differential expression analysis.

Response: N/A indicates no RPKM calculated for this gene. We revised the title of Table S11. A total of 7 enzymes (see Fig. 5d) encoding by 23 genes (see Table S11) in the anthocyanin biosynthesis pathway. The previous description would lead to

misunderstanding and we revised it in this version.

L323-324: "Thus, regulation of these... " The logic is misled. No valid explanation has been provided.

Response: We deleted this sentence.

L328-329: what do you mean "due to large number of these genes in the genome"? Please clarify.

Response: It should be "due to a large number of genes in this gene family in the genome". We revised this description.

L330: P-value or P-adjust?

Response: It should be "false discovery rate", we revised it.

L376: In Methods part, the authors should provide the version and citation of the programs and softwares. Detailed methods or pipeline are required, such as genome size estimation, the model of sequencing platform.

Response: We already added this information. Some software has no citations and we provided the related websites that the software can be downloaded.

L393: Which model of the sequencing platform? Novaseq6000 or others?

Response: Novaseq6000, we added this information.

L412: BUSCO, which lib was used?

Response: We added this information "The lineage dataset: embryophyta_odb10, Creation date: 2020-09-10, number of species: 50, number of BUSCOs: 1614".

L414: Did the clean hic reads are valid reads? How to generate the valid reads? Please provided more detailed information.

Response: Yes, the clean reads were valid. We used HiCUP to perform this analysis and we already added this information.

Fig1f, the authors should provide x-axis and y-axis information. Fig1g, complete the legend.

Response: The chromosome orders for x and y axis are the same and they are labeled in the bottom of this figure.

Fig2c, Required detailed explanation. For example, the means of words in red, green and blue. Fig2b,d, the authors should mark the p-adjust values.

Response: We revised this figure and its legend.

Fig4c, the authors should provide group information in the heatmap x-axis.

Response: We already provided this information, please see the bottom of Fig. 4c, "leaf S1 S2 S3 S4".

Fig. 5d, bHLH or BHLH?

Response: It should be "bHLH", we already revised it.

Reviewer #2 (Remarks to the Author):

This is an interesting study, and I look forward to seeing the revision. However, there are some significant issues with the methods and analyses as noted in the marked up manuscript. The major one is the likelihood that the true haploid genome of the tetraploid (44 chromosomes) was collapsed into a combined consensus (ancestral??) genome with only 22 chromosomes. Thus, the analyses are not accurate. The genome annotation and subsequent clustering and phylogenetic analysis would also benefit from

a more euphorbia-centric approach. As it is, the tree that was generated lacks statistical significance. Also, there are significant gaps in the methods that need to be addressed. On a personal note, I am also curious where leafy spurge fits into the comparisons (<https://www.cambridge.org/core/journals/weed-science/article/abs/gene-space-and-transcriptome-assemblies-of-leafy-spurge-euphorbia-esula-identify-promoter-sequences-repetitive-elements-highquality-markers-and-a-fulllength-chloroplast-genome/95C85C6B07094B6FA308C00608FDB816>) - particularly since it is also an invasive species in the US.

Response: First of all, thank you very much for your very constructive comments and suggestions to our manuscript. These comments and suggestions help us to improve the manuscript thoroughly and greatly.

Concerning to the question of genome size, I think the reviewer would misunderstand our result. The estimated tetraploid genome size of the CTT was 2.95 Gb by K-mer estimation. The genome sizes of the two CTT lines were estimated as 2934 and 3010 Mb by flow cytometry (Table S2). The genome of the CTT was then assembled by using two different software packages, Canu and HifiAsam. Both software programs produced a total of ca. 2.9 Gb contigs. The 2.9 Gb contigs were then purged into 787 Mb and a total of 22 large scaffolds (>10 Mb, most probably they are chromosomes) were clustered by using Hi-C data. Therefore, the haploid size of the CTT is 787 Mb and the tetraploid genome size is ca. 2.95 Mb. In the haploid genome, there are 22 chromosomes; while there are 88 chromosomes in the tetraploid genome. In the Flora of North America (http://floranorthamerica.org/Triadica_sebifera), The CTT also is reported to have 88 chromosomes. In the construction of phylogenetic tree, the haploid genomes were used for all plants. Moreover, we were not able to create the high-quality chromosome-scale genome for the CTT diploid (44 chromosomes) and tetraploid (88 chromosomes) genomes. Therefore, we think the haploid genome of CTT used for all analysis would be more suitable. We pointed out that the monoploid of CTT was used for analysis in most places.

Concerning to the lacks of statistical significance, the bootstrap values of all branches in the tree were added. The unigenes of leafy spurge were produced by assembling RNA-Seq and EST, thus, it is difficult to use this data to perform genome comparisons. Moreover, the editor and one reviewer suggested us to reduce the manuscript's emphasis on invasiveness because the plant material used for genome sequencing not collected from an invasive condition. However, we cited this literature in the introduction section. Finally, we expanded all the descriptions of our methods in this version.

Reviewer #3 (Remarks to the Author):

Manuscript COMMSBIO-22-0922, titled "The chromosome-scale genome sequence of the Chinese tallow tree provides insight into its polyploidization, high adaptability and biosynthesis of its high content of unsaturated fatty acids in seeds and anthocyanin in leaves" explores the genome of the Chinese tallow tree, describes its importance in Chinese culture - as a source of bioenergy and bioproducts, as an ornamental, and as a highly adaptable species. Based on the data presented, the authors conclude that the

Chinese tallow tree, a member of the Euphorbiaceae family, is a tetraploid likely resulting from whole-genome duplications, which increased genes for resistance, nutrition and energy utilization – thus, contributing to its invasive nature. The authors also include data to characterize genes and potential pathways associated with seed and leaf fatty acids and leaf anthocyanins.

Response: First of all, thank you very much for your very constructive comments and suggestions to our manuscript. These comments and suggestions help us to improve the manuscript thoroughly and greatly.

Overall, the manuscript is well written but the results section, particularly regarding the tetraploid nature, is a little wordy. Perhaps the manuscript could be more focused on the aspects of fatty acids and anthocyanins and less on the potential mechanisms of invasiveness. Because the genome sequence reported for Chinese tallow tree was from its native environment and not compared with the genome of its invasive counterpart in north America, it is pure speculation to suggest that the different categories of genes identified, compared with other species, is responsible for its invasiveness. Indeed, on Lines 339 – 342, the authors do acknowledge that the genome data presented is from Chinese tallow tree collected in China and not the invasive species of the U.S.A. Many other factors could be equally responsible for the invasive nature of Chinese tallow tree in north America. For example, *Euphorbia esula* (leafy spurge) is a perennial Euphorbiaceae species native to Eurasia that was accidentally introduced into north America, where it has become invasive. Numerous factors including an abundance of underground adventitious buds and root buds that go through well-defined phases of dormancy, and the lack of predatory pest common to its native range have been well documented as leading causes of invasiveness. The introduction of leafy spurge biocontrol agents from its native range, such as flea beetles, has helped to keep this invasive species in check – particularly in rangeland environments of north America. Thus, lines 342 – 347 “...the tetraploid CTT genome would be the major driver for its high adaptability and invasive character. Additionally, we also found that some genes related to disease resistance, nutrition and energy utilization, and secondary metabolite biosynthesis are specific for the CTT or expand significantly in its genome (Fig. 2, Tables S7 and S8). These genes would also be another driver for the tree’s high adaptability and invasive character.”, are again speculative in nature – as are the inclusion of Lines 102 – 106 suggesting the underlying mechanisms for invasiveness are due to “higher flavonoid concentrations in the roots and lower concentrations of tannins compared with native plant populations”. Certainly, the authors understand that Chinese tallow tree can become invasive; however, the data presented in this manuscript is too definitive on the mechanism(s) responsible. Without sequencing the genome of Chinese tallow tree from an invasive environment, or at least conducting an RNAseq study to determine if the genes in question from this study are overrepresented in invasive environments, the authors might consider reducing the manuscript’s emphasis on invasiveness and keeping it more focused on fatty acids and anthocyanins.

Response: Thanks for your comments. We totally agree with your suggestion. We already reduced the emphasis on invasiveness in this version.

Minor comments:

Line 67 – saying that “Numerous plants in this family can be used as bioenergy.” And then giving examples of products that are not associated with bioenergy (rubber and castor oil) is odd. Perhaps revise to “Numerous plants in this family can be used for generating bioenergy and bioproducts. For example...”

Response: We already revised all these points.

Line 293 – 294: “the RPKM>1 in at least one biological sample” this seems like a low level of stringency. An RPKM>2 across all biological replicates of a single treatment or time point for seed or leaves would be more appropriate.

Response: Because we used RPKM to perform this analysis, so RPKM>1 would be more suitable. Moreover, the data was just slightly changed if using RPKM>2 (only 3 more genes will be set as non-expressed genes).

Line 298: perhaps the authors could include a reference for “...in a previous report...”

Response: We changed the description of this sentence.

Reviewers' comments:

Reviewer #1 (Remarks to the Author):

The authors have revised my comments meticulously. I have no more comments.

Reviewer #2 (Remarks to the Author):

I did understand the the authors wanted to use the monoploid assembly because it produced chromosome sized contigs. However, I still maintain that the monoploid assembly is not likely the most accurate of the three assemblies that were developed. See my comments in the marked up manuscript at several points that support my suggestion that the Hifiasm assembly was likely the most accurate even though not all contigs could be assembled into chromosome-sized contigs. That said, if the authors acknowledge in the manuscript that the forced collapse of the two genomes into a single consensus assembly likely resulted in the loss of sequences that might differ between the two duplicated (paralogous) chromosomes - perhaps when presenting the BUSCO data or when noting that some contigs were that were covered by larger ones were discarded, that this would be acceptable. It might be a good idea though for the authors to check to make sure that the Hifiasm and the monoploid assemblies do not give too different a result when the genome is assessed for gene family expansion, or phylogentic analysis since the monoploid genome is clearly less complete than the Hifiasm assembly. It might also be a good idea to make both (if not all three) fasta files derived from the assemblies publicly available and downloadable from NCBI or some other public data repository and include the links in the manuscript if they are not already (it wasn't quite clear that these were all included in PRJNA813698).

Reviewer #3 (Remarks to the Author):

COMMSBIO-22-0922A - The chromosome-scale genome sequence of the Chinese tallow tree provides insight into its polyploidization, high adaptability and biosynthesis of its high content of unsaturated fatty acids in seeds and anthocyanin in leaves. The authors appear to have revised the manuscript to meet the concerns and suggestions of the reviewers. The manuscript is much improved and should be of interest to the scientific community. However, the authors should consider addressing the suggestions and comments listed below before final publication.

Line 88: suggest revising to read "In addition to being used for bioenergy production, the CTT..."

Line 129: "seeds of the CCT are coated with white wax (Fig. 1c)." However, Fig. 1c (page 39) shows red colored leaves. The authors should change to read "...seeds of the CCT are coated with white wax (Fig. 1d)." as is indicated in the figure legend of page 39.

Lines 145 – 148: on line 145 the purged size was listed as "ca. 787 Mb", on line 148 it says "778 Mb purged contigs" this seems confusing – should they both be "787 MB" as listed in Table S3?

Line 168: suggest replacing second "showed" in sentence with "had"

Line 184 – 185: this sentence indicates there are "...32,579 genes encoding 43,536 proteins..." yet, Page 32, Table 1 indicates "43,536 transcripts and CDS" and "32,579 Protein-coding gene". Suggest revising Table 1 to be consistent with line 184 – 185.

Line 266 -268: suggest revising to read "The seed of CTT produces stillingia oil inside the kernel and tallowy fat in the white coat outside the kernel (Fig. 4a)."

Line 269: why is it "Interesting"? The fact that one is an oil and the other is a fat should make it evident that the fatty acid composition is different.

Line 240: delete the second "that" in sentence to read "the fact that the CTT is a tetraploid would be..."

Line 359: just checking but the "779" does not seem to be consistent with what is reported elsewhere in manuscript?

Line 391: should it be "...that gene encoding" or "...that genes encoding"?

Line 396-398: suggest revising to read "Because Cy3-g is the major component of anthocyanins in red leaves of CTT, our results are consistent with studies reported in Arabidopsis." Also, maybe the authors should include a reference for this fact in Arabidopsis?

Line 402-403: revise to read "...showed increased expression due to a change in light (increased light/dark ratio) under cold conditions."

Line 142: has "HifiAsam" whereas line 463 has "HiFiasm" please be consistent.

Page 42: perhaps Cy3-g abbreviation should be spelled out in the legend and the abbreviation should be added in Figure 5c.

Reviewers' comments:

Reviewer #1 (Remarks to the Author):

The authors have revised my comments meticulously. I have no more comments.

Reviewer #2 (Remarks to the Author):

I did understand the authors wanted to use the monoploid assembly because it produced chromosome sized contigs. However, I still maintain that the monoploid assembly is not likely the most accurate of the three assemblies that were developed. See my comments in the marked up manuscript at several points that support my suggestion that the Hifiasm assembly was likely the most accurate even though not all contigs could be assembled into chromosome-sized contigs. That said, if the authors acknowledge in the manuscript that the forced collapse of the two genomes into a single consensus assembly likely resulted in the loss of sequences that might differ between the two duplicated (paralogous) chromosomes - perhaps when presenting the BUSCO data or when noting that some contigs were that were covered by larger ones were discarded, that this would be acceptable. It might be a good idea though for the authors to check to make sure that the Hifiasm and the monoploid assemblies do not give too different a result when the genome is assessed for gene family expansion, or phylogenetic analysis since the monoploid genome is clearly less complete than the Hifiasm assembly. It might also be a good idea to make both (if not all three) fasta files derived from the assemblies publicly available and downloadable from NCBI or some other public data repository and include the links in the manuscript if they are not already (it wasn't quite clear that these were all included in PRJNA813698).

Response: First, thank you very much for your suggestions and they help us to improve our manuscript greatly. We strongly agree with your comments. The monoploid genome would miss some sequence information. We followed your suggestion to compare the two genomes with a cluster analysis (See line 192 to 202). The results suggest that most gene in the diploid genome were represented by genes in the monoploid genome. Only 6.6% genes had no counterparts in the monoploid. We also acknowledged this limitation of the monoploid assembly in the section of discussion (See the second paragraph of 3.1 in the section of discussion). Additionally, it is difficult to use genes in diploid genome to perform analyses of gene expansions and phylogeny by using the diploid genome because the other 11 plants used in these 2 analyses are monoploid. Thus, we performed a similar cluster analysis just using the CTT diploid and monoploid genome sequence. We think this analysis would address your comments. We also deposited the diploid genome assembly to NCBI at 2022-06-18 and this data is also under the accession No. PRJNA813698.

Reviewer #3 (Remarks to the Author):

COMMSBIO-22-0922A - The chromosome-scale genome sequence of the Chinese tallow tree provides insight into its polyploidization, high adaptability and biosynthesis of its high content of unsaturated fatty acids in seeds and anthocyanin in leaves. The authors appear to have revised the manuscript to meet the concerns and suggestions of the reviewers. The manuscript is much improved and should be of interest to the scientific community. However, the authors should consider addressing the suggestions and comments listed below before final publication.

Response: Thank you very much for your suggestions and they help us to improve our manuscript greatly.

Line 88: suggest revising to read "In addition to being used for bioenergy production, the CTT..."

Response: Revised.

Line 129: “seeds of the CCT are coated with white wax (Fig. 1c).” However, Fig. 1c (page 39) shows red colored leaves. The authors should change to read “...seeds of the CCT are coated with white wax (Fig. 1d).” as is indicated in the figure legend of page 39.

Response: Revised.

Lines 145 – 148: on line 145 the purged size was listed as “ca. 787 Mb”, on line 148 it says “778 Mb purged contigs” this seems confusing – should they both be “787 MB” as listed in Table S3?

Response: It was typos, both are 787 Mb. It was revised.

Line 168: suggest replacing second “showed” in sentence with “had”.

Response: Revised.

Line 184 – 185: this sentence indicates there are “...32,579 genes encoding 43,536 proteins...” yet, Page 32, Table 1 indicates “43,536 transcripts and CDS” and “32,579 Protein-coding gene”. Suggest revising Table 1 to be consistent with line 184 – 185.

Response: Revised.

Line 266 -268: suggest revising to read “The seed of CTT produces stillingia oil inside the kernel and tallowy fat in the white coat outside the kernel (Fig. 4a).”

Response: Revised.

Line 269: why is it “Interesting”? The fact that one is an oil and the other is a fat should make it evident that the fatty acid composition is different.

Response: Revised.

Line 340: delete the second “that” in sentence to read “the fact that the CTT is a tetraploid would be...”

Response: Revised.

Line 359: just checking but the “779” does not seem to be consistent with what is reported elsewhere in manuscript?

Response: Revised. It should be 787.

Line 391: should it be “...that gene encoding” or “...that genes encoding”?

Response: Revised. It should be genes.

Line 396-398: suggest revising to read “Because Cy3-g is the major component of anthocyanins in red leaves of CTT, our results are consistent with studies reported in Arabidopsis.” Also, maybe the authors should include a reference for this fact in Arabidopsis?

Response: Revised. A reference was also added.

Line 402-403: revise to read “...showed increased expression due to a change in light (increased light/dark ratio) under cold conditions.”

Response: Revised.

Line 142: has “HiFiAsam” whereas line 463 has “HiFiasm” please be consistent.

Response: Revised.

Page 42: perhaps Cy3-g abbreviation should be spelled out in the legend and the abbreviation should be added in Figure 5c.

Response: Revised.

REVIEWERS' COMMENTS:

Reviewer #2 (Remarks to the Author):

Thank you for addressing my concerns about the monoploid verses the diploid genome assemblies. The changes made along with the submission of the diploid genome to the public database are a suitable compromise. I will endorse this manuscript.

Reviewer #2 (Remarks to the Author):

Thank you for addressing my concerns about the monoploid verses the diploid genome assemblies. The changes made along with the submission of the diploid genome to the public database are a suitable compromise. I will endorse this manuscript.

Response: Thank you very much for your suggestion and it helped us to improve our manuscript greatly.